# Potential Roles of Specific Subclasses of Premotor Interneurons in Spinal Cord Function Recovery after Traumatic Spinal Cord Injury in Adults

**DOI:** 10.3390/cells13080652

**Published:** 2024-04-09

**Authors:** Ana Dominguez-Bajo, Frédéric Clotman

**Affiliations:** Université catholique de Louvain, Louvain Institute of Biomolecular Science and Technology (LIBST), Animal Molecular and Cellular Biology Group (AMCB), Place Croix du Sud 4–5, 1348 Louvain la Neuve, Belgium

**Keywords:** adult spinal cord injury, interneuron, V2, sprouting, locomotor recovery

## Abstract

The differential expression of transcription factors during embryonic development has been selected as the main feature to define the specific subclasses of spinal interneurons. However, recent studies based on single-cell RNA sequencing and transcriptomic experiments suggest that this approach might not be appropriate in the adult spinal cord, where interneurons show overlapping expression profiles, especially in the ventral region. This constitutes a major challenge for the identification and direct targeting of specific populations that could be involved in locomotor recovery after a traumatic spinal cord injury in adults. Current experimental therapies, including electrical stimulation, training, pharmacological treatments, or cell implantation, that have resulted in improvements in locomotor behavior rely on the modulation of the activity and connectivity of interneurons located in the surroundings of the lesion core for the formation of detour circuits. However, very few publications clarify the specific identity of these cells. In this work, we review the studies where premotor interneurons were able to create new intraspinal circuits after different kinds of traumatic spinal cord injury, highlighting the difficulties encountered by researchers, to classify these populations.

## 1. Introduction

Traumatic spinal cord injury (TSCI) is a syndrome that affects thousands of people every year, mainly due to traffic accidents and falls during adulthood [1]. It substantially affects the quality of life of the patients, even reducing their lifespan, because of the apparition of severe motor, sensory, and autonomic disorders (e.g., respiratory, sexual, urinary, cardiovascular, and intestinal dysfunctions), depending on the level and the extent of the lesion [2,3]. In this review, we will focus principally on locomotor impairments. In the central nervous system, taking the spinal cord as a reference (which in its turn is divided rostro-caudally in the cervical, thoracic, lumbar, and sacral regions), we can classify the structures responsible for motor control as supraspinal centers and intraspinal circuits. Supraspinal centers (brain, brainstem, cerebellum) [4,5] send motor orders in a descending way through different tracts towards the spinal cord, the most studied being the corticospinal (CST), rubrospinal, tectospinal, and reticulospinal [6,7] tracts, but also through the gigantocellular formation, the cuneiform nucleus, the pedunculopontine nucleus [8,9], and the vestibular system [10,11] (Figure 1). On the other hand, intraspinal locomotor circuits are distributed all along the spinal cord. For example, the more relevant ones for the locomotor rhythm generation (known as rhythm generators) are mainly located in the ventral horns (laminae VII, VIII, X) of the lower thoracic and upper lumbar spinal cord (that is, within both the right and left hemicords of the lumbar enlargement, controlling both left and right hindlimbs), and they are also present in the cervical enlargement [12,13]. The main components of intraspinal locomotor circuits are motor neurons (MNs), which receive inputs from supraspinal centers and sensory afferents (in a direct or indirect way) and send their output to muscles and premotor interneurons (INs). Premotor INs, which have both their soma and prolongations inside the spinal cord, can pattern the final motor output either by connecting directly to MNs or indirectly through connections with other INs. Within premotor INs, propriospinal INs connect at least two spinal segments that could be either close to each other or even in the cervical and lumbar regions. Propriospinal INs are involved in the generation of motor reflexes, voluntary movement execution, sensory processing, and posture maintenance [14,15]. Some specific types of INs can be more abundant at one spinal level or another according to their function [16,17,18,19]. The production of normal locomotion also requires sensory inputs originating outside the spinal cord. In this case, sensory (touch, thermal, and nociceptive) information, along with proprioceptive information coming from the neural receptors present in our skin, muscles, joints, and other organs will travel through the sensorimotor afferents, which have their soma inside the dorsal root ganglia (DRG) and will enter the spinal cord through the dorsal horns to reach the intraspinal circuits (Figure 2). Once integrated inside the spinal cord, these inputs will modulate the motor commands and will in parallel be directed towards the supraspinal centers in an ascending way throughout the lemniscal, anterolateral, and spinocerebellar tracts. Hence, the connection between supraspinal centers and the spinal cord is disrupted after a TSCI, while spared spinal circuits around the lesion could still receive signals from the sensory receptors [20]. Interestingly, it was demonstrated decades ago that intraspinal circuits can generate locomotion, as well as the motor reflexes, independently of the supraspinal tracts [21,22]. In this review, we will focus on the premotor INs contained in motor circuits and how they react after different kinds of TSCI. For a broader information about the spinal circuitry responsible for the locomotion in the ventral spinal cord and central pattern generators, the authors strongly recommend reviewing the indicated publications [23,24,25,26,27,28].
Figure 1Schemes representing the location of the major white tracts in the human (**top**: cervical region as an example) or mouse (**bottom**) spinal cords. Colors of the white matter tracts of the mouse spinal cord follow the legend on the top left part of the image, according to the human spinal cord scheme. CC, central canal. Adapted with permission from [29].
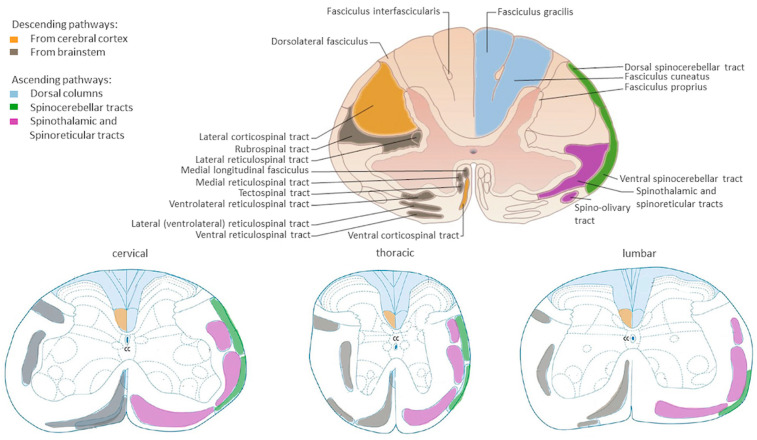

Figure 2**Left**: Schemes of propriospinal INs having their cell body at the cervical (green), thoracic (red) or lumbar levels (blue) of the spinal cord (reproduced from [27]). **Right**: Schematic representation of sensory afferents which their soma inside the DRGs, transmitting the information to INs and MNs through the dorsal horn. Roman numbers refer to the Rexed laminae (adapted from [30]).
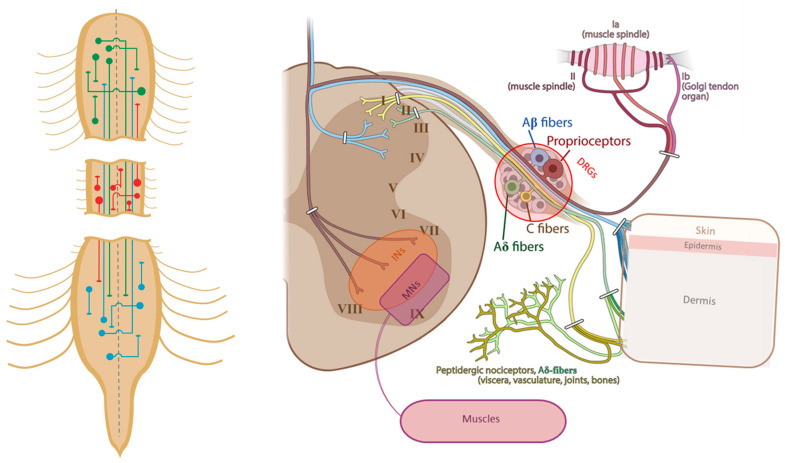



## 2. Current Challenges to Characterize Spinal Premotor Interneuron Populations in the Adult Ventral Spinal Cord

Premotor INs can be primarily classified following general neuronal features such as where they project (ipsilateral, if they project to the same hemicord where their soma is located, or commissural, when they cross the midline), if their axon is contained in the same spinal segment or they send their prolongations to other spinal levels (short or long), which neurotransmitter(s) they release (excitatory or inhibitory), and their firing properties [12,15,31,32]. However, there is still no consensus about the best way to group different subsets of INs in a meaningful manner to be able to study them separately, as well as to describe the interrelations among them, in the adult ventral spinal cord [25,33,34]. At present, the most used classification has been the cardinal system, in which cardinal cell populations are named according to their progenitor cells and the expression of specific transcription factors during the early embryological stages [35]. For instance, the ventral neural tube contains 5 interneuron progenitor domains (pd6, p0, p1, p2 and p3) that produce different interneuron populations (dI6, V0, V1, V2 and V3). These cardinal classes of INs have been further divided into main populations characterized by unique markers, like Chx10 for V2a, Gata3 for V2b, Sox1 for V2c, and Shox2 for V2d Ins (Figure 3). V0d, V1, and V2b INs are GABAergic/glycinergic; on the other hand, V0v, V2a, V3, and Hb9-INs are glutamatergic, and V0c INs are cholinergic, while neurotransmitters have not been identified yet for the other subsets (reviewed among others in [36]. This classification has helped researchers to isolate these big clusters of neurons at early stages of development to properly study their contribution to the locomotor process [23,24,25,26,27,28]. However, subsets of these original main populations have been identified over time, based on the expression of other molecular markers, whose functions are still unknown [37,38]. Moreover, molecular definition of cardinal neuron subtypes becomes less robust with time in ventral regions than it does in dorsal regions at postnatal stages. That is, dorsal INs become highly distinct over development as ventral INs present overlapping and shared molecular markers in the adult spinal cord [34]. On top of that, the expression of the embryonic markers varies according to the developmental stage and the spinal level of interest. Finally, identical neurotransmitters are used by multiple IN subsets, and the neurotransmitter environment can change after TSCI [39]. Taken together, these constitute an obstacle for researchers working in spinal cord disorders: first, in gaining insight into the general architecture and functioning of the adult spinal cord, and secondly, because in the TSCI context, they make it quite difficult to describe which specific subset of spared INs (in the surroundings of the injured area) could be involved in the potential reparation of the spinal cord after a traumatic event. For example, recent outstanding work performed by Hayashi and colleagues [17] in mice described for the first time the different expression patterns of transcription factors in V2a Chx10+ (Vsx2+) INs from the embryological to the postnatal stages along the whole spinal cord. They demonstrated that V2a INs are composed of 2 subsets that differ in their temporal expression of *Chx10* transcription factor, their abundance depending on the spinal level (cervical versus lumbar) and their projection targets (supraspinal centers versus neighboring neurons), suggesting that they are present in different motor circuits across the spinal cord. Moreover, Shox2, which is normally used to identify V2d INs, has been also reported as a marker of cervical spinocerebellar neurons, but cannot be used in the lumbar region to identify these cells [18]. Furthermore, in the case of V3 INs, it has been demonstrated that spatial distribution of V3 subsets is linked to their differential neuronal activity, while cell morphology relates to the temporal expression of specific transcription factors, leading to many V3 subtypes [32]. In the last few years, great effort has been expended to refine IN clustering by pooling the results of different RNAseq/transcriptomics experiments obtained by many research groups [33,34,40,41]. However, there is a general shared concern about the extreme difficulty of identifying postnatal ventral cell types. It has been hypothesized that, as the cell populations located in the ventral horns belong to a more heterogeneous circuitry than the cells located in the dorsal horns, the transcription factors defining them during the early stages of their differentiation are more prone to be downregulated over time, to permit the expression of other factors more related to their final postnatal function [34]. Methods complementary to the cardinal classification, such as clustering according to cell birth time or location, are being proposed to improve the ways of grouping INs. Indeed, recent evidence has demonstrated that cardinal categorization does not consider all the described locomotor rhythm-generating neurons (e.g., V2d and the Hb9 expressing INs) [25]. Thus, new databases are being generated to merge all the knowledge coming from different studies and to relate the spinal information coming from early, young, or adult individuals of different species to help translational research potentially applied to spinal cord injury [42,43].
Figure 3**Left**: Eleven IN progenitor domains produce different IN populations (ventral ones correspond to dI6, V0, V1, V2, and V3). **Right**: comparison between IN connectivity between zebrafish and mouse. Adapted from [44].
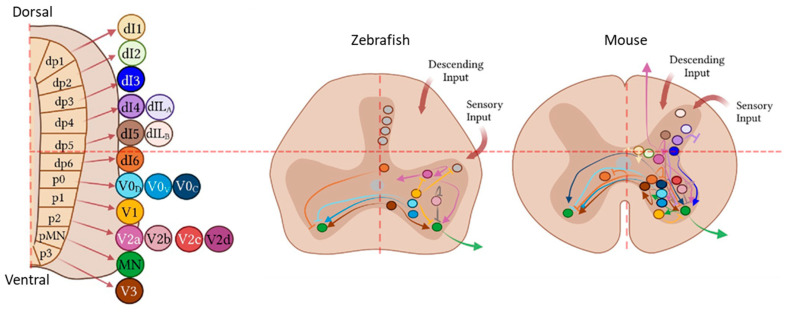



## 3. Limitations in Defining the Potential Roles of Interneurons for Spinal Tissue Recovery after a TSCI

The most popular publications describing the potential roles of INs in spinal tissue recovery after a TSCI date from the late 1990s to early 2000s; in them, it was reported that spared supraspinal projections (in this case belonging to the corticospinal tract) that were injured due to a traumatic event were able to create new collaterals through a process called “sprouting”, in order to form new synapses with spinal INs around the lesion site as an attempt to bypass the damage [6,45,46]. These signs of intraspinal tissue reorganization seem to be evident from three weeks after injury [47,48]. In thoracic incomplete lesions in adult animals, INs located in the segment just above the lesion can undergo plasticity processes at both the dendrite and axonal levels. For example, they can increase the number of proximal dendrites, the quantity of total dendrites that cross the midline, and the number of dendritic spines. In fact, hints of sprouting such as growth cone-like endings in these new branches were detected more than 4 months after lesion. Interestingly, injured axons create collateral branches in both ascending and descending directions, which could lead to changes in the original spinal circuitry [49]. However, in this last example as well as in many other works related to adult spinal tissue, INs were not classified as belonging to any of the cardinal subclasses. Additionally, in some cases, INs are described as being “all the neuronal cells contained in the spinal cord that are not MNs” (which are clearly distinguishable thanks to their morphological features), and detected at different time points after injury in the surroundings of the lesion site, or inside the lesion core, by general neuronal markers such as β-III-Tubulin, Tau, MAP2, NeuN, or SMI311, instead of population-specific labeling [49,50,51,52]. This is not surprising since, as we still do not know which molecular markers could be specific of ventral adult populations, there is a lack of antibodies or probes to classify them, unless transgenic zebrafish or mouse lines created by using the genetic markers described for embryonic lineage populations are used to globally follow the early defined groups of INs [22]. On top of that, there is also no consensus about (1) how to compare the results regarding propriospinal IN reorganization when different TSCI models are applied (transection, hemisection, or contusion with different injury sizes), (2) whether hints of neurite sprouting can be compared when the lesion occurs at different levels (cervical, thoracic, or lumbar), and (3) how to relate the results when different animal experimental models are used (although increasing comparative anatomy studies about the non-injured spinal cord in different species have been published [44,53]) (Figure 3). Moreover, because of technical limitations, it is not always possible to confirm that the neuronal components that are bridging the lesion site belong to active neurons, nor that they re-establish, at least to some extent, the original connections that exist in the normal non-injured spinal cord. The second and third point are related to what it is known as the neuroanatomical–functional paradox after a TSCI [54]. It has been repeatedly shown that lesions with a similar extent, at the same level, where signs of neuronal sprouting have been detected (both with supraspinal projections or between different kinds of INs), could lead to very different functional consequences in the case of injured cervical and/or thoracic tissue. Regarding animal models, in species with regenerative capacities such as zebrafish, motor restoration can happen with the regeneration of just one-third of the total amount of supraspinal descending axons and intraspinal long-projecting INs (connecting the cervical and lumbar levels) present in healthy animals [55]. In small mammals (mice and rats), it has also been described that a very small percentage of fiber recovery (around 2–7%) could be enough to obtain motor improvements [56]. In adult humans, it is well known that the central nervous system healing capacity is very limited, even if many patients living with chronic TSCI show some degree of spontaneous sensorimotor recovery below the initial spinal injury level. When this is the case, it usually happens during the first nine months after the accident, reaching a plateau around a year post-injury. Importantly, the recovery rates are better in the case of patients with incomplete lesions, thanks to the amount of preserved functional nervous tissue that can be used as a relay to bridge the lesion. Furthermore, the older a patient is, the more difficult the recovery will be, and there is currently not enough data to assess whether gender could have an impact on facing the symptoms. Also, conversion from a complete to an incomplete injury is more common in tetraplegia than paraplegia [3]. All this evidence suggests that there are still essential restorative mechanisms that need to be identified for researchers and clinicians to be able to develop better therapies [19]. However, it is critical to keep in mind that this recovery is not only dependent on neuronal damage, but also to the attenuation and stabilization of other body responses such as the immune response, vascular damage, and mechanical stabilization of the tissue, among others. Thus, gain of movement is not dependent only on the formation of axonal projections able to cross the lesion gap [2,5,57].

## 4. Ventral Interneuron Contribution to Spinal Tissue Restoration in Different TSCI Models

Most of the research community working in TSCI has been focused either on targeting projections from supraspinal centers (mainly from the corticospinal tract), which may create new synapses with spinal INs [58,59], or on propriospinal IN circuitry modulation all along the spinal cord, because it has been demonstrated that spinal tissue reorganization can also occur below the lesion. Small mammals and zebrafish are the main experimental animal model used for IN characterization, followed by non-human primate models. 

### 4.1. Cervical TSCI

Major attention has been put on studying TSCI when it occurs at the cervical level, particularly its effects on upper limb motor activity and respiratory circuitry [60]. For example, it seems that propriospinal neurons located in the C3-C4 cervical segments can relay cortical signals after a TSCI to restore hand dexterity in non-human primate (NHP) models. Interestingly, these cells are not essential in uninjured individuals but become indispensable for the recovery process [61], as happens with V2a INs at mid-thoracic level [19]. Cervical (C6) right hemisection in adult rats also leads to impairments in ipsilateral forelimb mobility. Injured animals have received the implantation of a biocompatible scaffold placed at the lesion site as an attempt to help spinal tissue regeneration. A substantial number of newly grown vGlut2+ neurites were found at the lesion area four months after the surgery, leading to the hypothesis of a potential reorganization of preexisting reticulo-propriospinal connections coming from the brainstem, even though there were no significant motor improvements at that time [50]. In an NHP model of incomplete cervical hemisection, electrical stimulation of dorsal roots at low cervical levels of the spinal cord during reach or grasp/pull movement phases improved arm movements, likely evoked by residual descending inputs, but the involved neuronal populations remained uncharacterized [62]. Regarding breathing alterations, works have been mainly focused on how phrenic MNs, which control the movement of the diaphragm, are affected and thus on potential changes in pre-phrenic INs located in C3–C6 segments [63]. For example, implanted grafts of human neuronal stem cells derived from the H9-NSC line combined with neurotrophic factors (BDNFs) have shown good integration into the spinal tissue when applied to cervical (C5) hemisected rat spinal cords, but the possible formation of new synapses with specific classes of interneurons was not investigated [51]. In mammals, the recruitment of developmentally defined V2a neurons during the recovery of walking and breathing suggests that these cells modify their interactions with the circuits that control these functions [61]. After demonstrating that V2a INs reorganize in the cervical tissue after a C2 hemisection [64], Zholudeva and colleagues transplanted neural progenitor cells enriched with iPSC-derived V2a after a lateralized contusion injury in the area comprising C3-C4 spinal segments. V2a-like cells contributed to diaphragm functional improvement one month after treatment [65]. Moreover, it has been shown that expiratory bulbospinal neurons boost their connections with propriospinal INs (but not MNs) in the segments above the lesion after a thoracic lesion [66]. Interestingly, neurons ipsilateral to an upper cervical hemisection (C2) do not present the same activity as those located in the contralateral hemicord, and these changes in neuronal activity can be modulated in hypoxic conditions [67]. Thus, spontaneous circuit reorganization and therapeutic interventions can collaborate to promote motor and respiratory recovery. Moreover, after a high incomplete cervical lesion, sprouting processes happening in affected ascending proprioceptive axons coming from the ipsilateral cervical DRGs, enable these afferents to create new connections in the grey matter with cuneate-projecting INs to boost tactile and proprioceptive function recovery [68]. Afterwards, Courtine’s lab demonstrated that epidural electrical stimulation could be useful for functional recovery even when applied after a cervical lesion. A device targeting the dorsal root entry zones of the lumbosacral spinal cord was implanted in a patient with a chronic TSCI due to a cycling accident (incomplete cervical lesion at C5/C6 levels). Importantly, the patient was able to calibrate the device himself without extra medical assistance after the first guided manipulations. After one year of treatment combined with training routines, it allowed him to walk with crutches [69]. Currently, most influential methodologies are based on the electrical stimulation of the dorsal roots coming from the DRGs at caudal regions of the spinal cord [57]. Epidural electrical stimulation supported by training to treat injured animals or humans does not directly target spinal premotor INs, but modulates the activity of proprioceptive and sensory afferents, that in turn, will impact the activity of INs and stimulate neuronal circuit reorganization [55]. Thus, when supraspinal projections are compromised, sensory information could become the main source of control for movement [20,61]. This technology has been also applied after thoracic injuries, which corresponds to the level of choice in the majority of TSCI research approaches. 

### 4.2. Thoracic TSCI

Locomotor activity was improved in mice suffering from a mid-thoracic contusion after the electrical stimulation of the ensemble of thoracic, lumbar, and sacral dorsal roots placed below the lesion, which was applied in combination with rehabilitation training [57]. It was demonstrated that the main actors in locomotor improvements were a subset of V2a cells (Hoxa10+Chx10+) located in the intermediate lamina of the ventral lumbar spinal cord. These cells receive inputs from both reticulospinal neurons coming from the ventral gigantocellular nucleus and from large-diameter sensory afferents of parvalbumin+ neurons located in the DRGs, while they exclusively project to glutamatergic, gabaergic, and cholinergic neurons located in the ventral spinal cord. Optogenetic inactivation of lumbar V2a in mice that suffered from TSCI instantly suppressed walking previously enabled by electrical stimulation, while it was immediately restored if IN activity was not disrupted. In fact, animals lacking Chx10+Hoxa10+ INs that suffered from a unilateral lesion failed to fully recover locomotion [54]. Electrical stimulation was also applied in a subpopulation of V2a INs (long-propriospinal ZFH+ INs) connecting the cervical to the lumbar cord to boost tissue restoration after a mid-thoracic injury (T8) [58].

Additionally, the detailed characterization of gait types in uninjured and injured animals (rats and cats) enabled a more refined description of the reorganization of long propriospinal INs after injury, showing that it may not happen in the same way when two different kinds of traumatic lesions (hemisection or contusion) are applied at the same thoracic level (T10) [13,70]. After suffering a hemisection, animals used the side contralateral to the damage to lead their movement, but their locomotor performance at slow speeds was not that different from the one without injury. This could relate to the fact that long connections could be reinforced in the preserved hemicord, and that this non-injured spinal tissue could support neuronal sprouting happening in the surroundings of the injured area as an attempt to bypass the damage. This would be associated with the fact that interlimb coupling is better preserved after hemisection than after a contusion affecting both sides of the spinal cord, which results in more affected motor performance in general, and even to the appearance of novel mobility patterns [13]. However, preservation of long-connecting fibers is not beneficial in all contexts regarding functional recovery. Long ascending propriospinal INs (LAPNs) at L2 connect the lumbar and the cervical enlargements. Given the ventrolateral location of LAPNs in the spinal cord white matter, they are more likely to remain uninjured after SCI, so in principle, silencing them after a thoracic TSCI (T9) could disrupt locomotor recovery. However, silencing spared LAPNs’ post-SCI improved locomotor function, although the reason for this remains unclear. One possible explanation could be that silencing these LAPNs prevents maladaptive plasticity with sensory afferents, as both proprioceptive and nociceptive afferents caudal to the injury are able to sprout post injury [71].

Regarding the supposed beneficial role of long propriospinal IN reorganization, it is also controversial that injured animals are able to move their hindlimbs even after two spared-in-time hemisections in contralateral hemicords at different thoracic levels (i.e., the first one happening more rostrally, followed by a second one at a contralateral caudal level), where all the long ascending and descending connections are therefore impaired. In fact, both left and right spinal circuitry within the lumbar spinal cord remain active even if these animals suffer a third injury consisting of a complete sectioning of both hemicords at a more caudal level two months after the second hemisection. This may result from previous lumbar spinal tissue reorganization after the two first hemisections (comprising short propiospinal connections), along with the integration of sensory feedback coming from both hindlimbs [70].

Furthermore, after TSCI at the thoracic level, de novo collaterals belonging to spared transected supraspinal tracts that try to bypass the lesion do not always find the correct rewiring paths. To overcome this issue, the controlled delivery of molecules involved in synapse formation have been applied in the surroundings of the injury. For instance, the overexpression of FGF22 in long propriospinal INs in the cervical spinal cord on the first day after injury (T8) reinforces the newly created CST-IN synapses by increasing the formation of excitatory contacts onto them, as well as increasing motor neuron survival in the lumbar cord [59]. With the same idea, the controlled delivery of neurotrophic factors below the lesion area led to a substantial recovery of walking after a complete low thoracic injury in mice, where animals showed a gait similar to unilateral injured animals [19]. In this case, the applied molecules (a cocktail of growth and neurotrophic factors) were selected, inspired by embryonic stages when spinal neuronal circuits are built, to mimic developmental conditions. Most of the involved neurons corresponded to a long-distance projection subtype of V2a Ins (Hoxa7+Chx10+Zfhx3+), whose cell bodies are in the mid-thoracic spinal cord, but their axons reach the lumbar part. These results were confirmed through the fact that silencing regenerated Zfhx3+ neurons disrupted the regained improvements in locomotor activity after treatment. Interestingly, it has been demonstrated that the ablation of mid-thoracic INs (including Chx10+ INs) has no detectable impact in locomotion in uninjured adult mice, but according to the results shown and in concordance with additional works [19,57], there is increasing evidence that they become key mediators for function recovery, especially after unilateral TSCI. They also found that stimulation of the ventral gigantocellular nucleus induced large motor evoked potentials in leg muscles, revealing that supraspinal centers regained functional access to the lumbar spinal cord thanks to INs reorganization. Finally, locomotor improvements were detected in rats, suffering from a complete transection (T8-T9), that followed a long-lasting training routine combined with a pharmacological treatment based on a 5-HT agonist applied 9 weeks after the beginning of exercise [72]. The authors suggest that this therapy prompts the reorganization of premotor IN connectivity to MNs controlling the gastrocnemius and tibilialis anterior located at L4-L5 lumbar levels. Such circuit reorganization could be promoted by dynamic changes in gene expression, resulting in an increased transcription of genes related to axon guidance and regeneration [18].

Additionally, it seems that complete lower thoracic TSCIs in adults promote the conversion of excitatory V2 INs (Shox2+) into inhibitory INs in the ventral spinal cord, thus inhibiting instead of activating MNs. Thus, preventing this phenotype switching by implementing training routines helped to obtain better locomotor outcomes after injury. This neurotransmitter switch did not happen in V3 INs that receive fewer connections from propriospinal afferents than V2. Thus, it seems that the maintenance of the original glutamatergic circuitry is also dependent on the proper functioning of propriospinal afferents and in the number of connections formed with them [39]. In concordance with previous results, detrusor–sphincter impairments that appear in lower-transected rats could be also due to alterations both in excitatory IN activity and in dorsal root afferents happening at the lumbosacral level after injury [73]. Moreover, Shox2 IN glutamatergic/inhibitory inputs are also dysregulated after complete transection at T8-T10 thoracic levels. Although their intrinsic activity is not significantly affected, they become more sensitive to serotonin [74]. These results are in concordance with the hypersensitivity to serotonin that V2a cells located in the lumbar spinal cord presented after a complete T8-T9 transection in adult mice [75]. Other kind of INs, such as dI4 (derived from the differentiation of a hESC cell line in vitro) have been implanted in the lower thoracic hemisected spinal cord, where they survived and integrated into the surroundings of the injured tissue for up to 7 months in a nonhuman spinal cord injury model [76]. Which phenotypic changes occur in these transplanted neuronal populations after transplantation and during integration into the preserved circuits remains in most cases to be investigated. Work in zebrafish demonstrated that V2a (Chx10+) INs generated de novo from endogenous neural progenitors nine days post-injury were able to survive and synapse with motor neurons located far from the lesion site both in larvae [77] and in adult fish. Interestingly, in postnatal stages, V2a contributes to bypassing the lesion site following a subtype specific chronological order [5]. 

### 4.3. Lumbar TSCIs

Regarding the lumbar level, it has been demonstrated that the upper lumbar cord is more relevant to locomotor function than the lower lumbar or lower thoracic levels. Thus, interest about the potential role of the spared long-ascending propriospinal interneurons, that have their soma located at this region and project to lower cervical levels in locomotor recovery has been increasing [13]. V3 propriospinal INs are implicated in spasm generation after a complete transection at the sacral level. Interestingly, in this case, cells did not suffer from neurotransmitter phenotype switching [78], as happens with V2 after a thoracic lesion [39]. Usually, spasm apparition is described because of maladaptive sprouting (V3 creating aberrant connections that are not present in the healthy tissue with other INs and MNs), but Lin and colleagues proposed that they should be also studied as an initial point for mobility restoration, considering that spasms could be a manifestation of partially recovered function in the spinal central pattern generators. Taken together, these observations suggest that a reinforcement of proprioceptive signaling to intraspinal circuits located below the lesion site in the absence of normal descending connectivity could be a key determinant modulator of the formation of detour circuits for spontaneous locomotor recovery [20].

## 5. Conclusions, Limitations, and Perspectives regarding the Study of the Role of Interneuron in Intraspinal Tissue Remodeling after a TSCI

Currently, there is no agreement about the best way to cluster or name all the populations and subpopulations of premotor INs in the adult ventral spinal cord. It is known that the expression of specific transcription factors at well-defined developmental stages will, at least in part, set their identity and role in the spinal circuitry. For example, it is possible to predict whether a cell would project to distal levels (even to supraspinal centers) or connect locally according to a well-defined combination of active transcription factors at early stages of embryonic development. Conversely, that information is not sufficient to predict their postsynaptic targets, or the neurotransmitter that they will release [33]. This lack of specific markers in the postnatal stages means that, for the moment, researchers remain with the need of generating complex transgenic animals in which the tracing of every IN subpopulation should be designed under the control of transcription factors present at embryonic stages. This implies verifying that all kind of INs belonging to the studied population will be labeled throughout the life of the animals (that is, in the postnatal stages of interest). On the other hand, the use of lineage tracing where the conditional activation of a reporter gene to visualize a specific IN subset in adulthood is dependent on drug administration during development (e.g., tamoxifen) could lead to undesired abortions [79]. That scenario could be at least partially solved with a broader knowledge of transcription factors or other markers that define/characterize these IN subpopulations at postnatal stages. Additionally, the generation of specific mouse transgenic lines for every IN subset would not be either economically or temporally feasible [25]. For all these reasons, it has been hypothesized that it possibly does not make sense to classify these cells in clusters in the adult spinal cord [34]. 

In the context of TSCI, it seems that a diversity of mechanisms involved in spinal tissue restoration have not been discovered yet, because, even in the cases where multidisciplinary approaches have been used, results regarding spinal tissue restoration remain puzzling. The sprouting of supraspinal axons to create new durable connections with spinal INs [58,59], along with propriospinal interneurons reorganization both allowing the reconnection of distal parts of the spinal cord after lesion or at a more local level, are crucial for locomotor recovery. Regarding intraspinal circuits, V2a INs are frequently identified, either autonomously or in response to therapeutic interventions, as contributing to functional recovery after TSCI (refs of V2a-related papers). However, this seems due to the higher abundance of markers and tools to identify these cells, including the persistence of Chx10 in a subset of them [17], rather than to a specific propensity of V2a for axonal reorganization or response to stimuli, as other Ins display similar organization or function in spinal motor circuits [23,24,25,26,27,28]. Accordingly, advances in IN characterization highlighted other neurons which belong to supraspinal centers and share some genetic features with propriospinal INs, and whose activation impacts upon the activity of ventral horn neurons [9]. Thus, a better understanding of the identity of spared INs able to sprout and to reconstitute functional circuits in the different injury models at different levels of the spinal cord will be critical [80] to design better therapies to support their survival and to boost their growth and maintenance for the recovery of spinal tissue functionality [19]. In parallel, being able to isolate and silence the INs forming aberrant connections that result in detrimental outcomes and poorer life quality to the patients, such as the appearance of pain, spasms, or impaired mobility amelioration due to maladaptive adaptations, would also be of great importance [71]. Finally, one group recently suggested that, after a cervical TSCI, injured but spared axons coming from supraspinal centers, such as the cortico-spinal tract, could somehow revert to an embryonic transcriptional state, which would enable them to create collateral neurites for the regeneration of the damaged nervous tissue [81]. If in the coming years it is demonstrated that this reversion also applies to spared premotor INs, along with a better understanding of which subtypes are implicated in this process, this will open great path for the development of more directed and efficient translational strategies for the recovery of locomotor activity.

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
