# Peer review of "Potential Roles of Specific Subclasses of Premotor Interneurons in Spinal Cord Function Recovery after Traumatic Spinal Cord Injury in Adults"

_cells, 2024, doi:10.3390/cells13080652_

Round 1
Reviewer 1 Report
Comments and Suggestions for Authors
This a very good review and with good illustrations. The topic has been nicely covered and I contratulate the authors for this work.
The wording can be confused for some readers. The authors use propriospinal neurons and premotor neurons as the same set of spinal interneurons, which is probably right. However, propriospinal neurons are also involved in reflexes, in addition to locomotion and posture. It would be convenient to define these names in order to increase clarity.
In line 32, note that "of the" seems an incomplete setence.
Reviewer 2 Report
Comments and Suggestions for Authors
This is an excellent review that summarizes studies related to ventral interneurons in the spinal cord and the changes in circuits that they form after spinal cord injury. The authors who are experts in this field also discuss the difficulties in defining sub-classes of interneurons given the complexity of these populations. Finally, they present a perspective on therapeutic strategies to restore locomotion function.
Although recent publications (such as Dougherty et al 2023) provided a similar review on spinal cord interneurons, this review adds important and valuable perspective. My suggestions are therefore focused on organization and presentations which in my view will help both expert and no-experts to follow the review.
1. The review is organized into different spinal cord levels, cervical, thoracic and lumbar. The lumbar part is weak despite the importance and rich information on the CPG circuits both in terms of experimental data and modeling. This section should be further developed with presentation of recent data and models (for example work from the Rybak lab) including a figure to demonstrate past controversies and current progress .
2. Table 1 is difficult to follow, particularly the small letter notations in the cervical/thoracic/lumbar presentations.
3. It would be helpful to add to one of the figures (and text) information about the neurotransmitters that define the interneurons and discuss how they may define the sub-populations.
